# Interpretable Neural PDE Solvers using Symbolic Frameworks

## Abstract

Partial differential equations (PDEs) are ubiquitous in the world around us, modelling phenomena from heat and sound to quantum systems. Recent advances in deep learning have resulted in the development of powerful neural solvers; however, while these methods have demonstrated state-of-the-art performance in both accuracy and computational efficiency, a significant challenge remains in their interpretability. Most existing methodologies prioritize predictive accuracy over clarity in the underlying mechanisms driving the model's decisions. Interpretability is crucial for trustworthiness and broader applicability, especially in scientific and engineering domains where neural PDE solvers might see the most impact. In this context, a notable gap in current research is the integration of symbolic frameworks (such as symbolic regression) into these solvers. Symbolic frameworks have the potential to distill complex neural operations into human-readable mathematical expressions, bridging the divide between black-box predictions and solutions.

## 1 Introduction

Partial differential equations (PDEs) are ubiquitous to modern physics and engineering. Though they have existed since the first system of equations found by Euler over 250 years ago, many challenges remain in finding numerical solutions resolving spatiotemporal features over multiple scales and nonlinear stochastic systems. Currently, numerical solvers such as finite differences and finite elements can solve PDEs when analytical solutions cannot be found, like in nonlinear and high dimensional systems, by discretizing a problem over a grid and evolving over time at very fine-grain timesteps. However, such solvers are computationally complex even with modern computers, and can fail when faced by multiscale and/or stochastic PDEs.

### 1.1 Neural PDE Solvers

Neural networks can learn and generalize to new contexts such as different initial/boundary conditions, coefficients, or even different PDEs entirely. They are a particularly promising avenue for solving highly complex PDEs such as those found in weather prediction and fluid dynamics. For a review which contextualizes physics informed machine learning with regards to classical problems and methods, see [22].

Today, neural PDE solvers are capable of remarkable performance in historically challenging tasks. FourCastNet, proposed by Pathak et al. [24], achieves comparable short-term forecasting precision to the ECMWF Integrated Forecasting System (IFS), a cutting-edge numerical weather prediction (NWP) model, for large-scale variables like atmospheric pressure. It surpasses IFS in predicting small-scale variables, such as precipitation. Most notably, however, is the fact that it is 4 to 5 orders of magnitude faster than most NWP models and uses a fraction of the number of variables to achieve

Submitted to NeurIPS 2021 AI for Science Workshop, Attention Track.

its remarkable performance. This success is based off of the popular neural operator framework and, more specifically, Fourier neural operators (FNOs) [18].

However, models designed to solve more fundamental mathematical tasks like PDEs directly do not see as much acclaim or practical adoption. While we use PDEs to model the mathematics underlying heat, sound, electrodynamics, fluid dynamics, and many other physical phenomena which are essential to many engineering and medical tasks (among other fields), one major issue hinders real-world applications of neural methods: interpretability.

Neural PDE solvers lack the guarantees and transparency that numerical solvers have. This prevents them to be used in what are often high-stakes applications like engineering and medical simulations, even despite any potential gains in accuracy, generalisability, or computation time.

Now that we have demonstrated the clear potential for cutting edge neural PDE solvers, some of which are laid out below, focus should move toward bringing these methods closer to real-world application by considering needs beginning with interpretability and extending to more stringent demands such as performance bounds or limits.

## 2 An overview of the current methods

The idea of using deep learning techniques to solve differential equations has a long history, including Dissanayake's and Phan-Thien's attempt to use multilayer perceptrons (MLPs) as universal approximators to solve PDEs, and arguably includes any work involving incorporating prior knowledge into models in general [9, 15, 25]. Although some hybrid classical-neural methods exist [13, 23, 31? ], which are inherently interpretable to varying degrees, we focus on fully neural methods: methods which rely on the universal function approximation theory, such that a sufficiently complex network can represent any arbitrary function. One very popular method mentioned already above is the neural operator, which models the solution of a PDE as an operator mapping inputs to outputs. The problem is set such that a neural operator $\mathcal{M}$ satisfies $\mathcal{M}(t, \mathbf{u}^0) = \mathbf{u}(t)$ where $\mathbf{u}$ is the solution, and $\mathbf{u}^0$ are the initial conditions [5, 20]. Simple MLPs, convolutional neural networks (CNNs), recurrent neural networks (RNNs), and other networks used to map input vectors to output vectors are naive examples of finite-dimensional operators.

### 2.1 Physics Informed Neural Networks

In 2017, Raissi et al. introduced the physics-informed neural network (PINN) [27]. They structure the problem such that the network, denoted as $\mathcal{N}$, fulfills $\mathcal{N}(t, \mathbf{u}^0) = \mathbf{u}(t)$, with $\mathbf{u}^0$ representing the initial conditions. The core idea of PINNs is to directly integrate relevant physical laws into the network's predictions, which is achieved by incorporating additional loss term(s) into the network's objective function.

For a typical loss function $\theta = \text{argmin}_\theta \mathcal{L}(\theta)$

the loss with a physics prior may be defined as follows:

$$\mathcal{L}(\theta) = \omega_\mathcal{F} \mathcal{L}_\mathcal{F}(\theta) + \omega_\mathcal{B} \mathcal{L}_\mathcal{B}(\theta) + \omega_d \mathcal{L}_{\text{data}}(\theta) \tag{1}$$

where $\mathcal{L}_\mathcal{B}$ penalises against the initial and/or boundary conditions to fit the known data over the network, $\mathcal{L}_\mathcal{F}$ enforces the PDE itself at collocation points (which are calculated using auto-differentiation to compute derivatives of $\hat{\mathbf{u}}_\theta(\mathbf{z})$), and $\mathcal{L}_{\text{data}}$ (the standard loss) forces $\hat{\mathbf{u}}_\theta$ to match measurements of $\mathbf{u}$ over the provided data points.

Since the network maps input variables to output variables which are both finite-dimensional and dependent on the grid used to discretize the problem domain, it is considered a finite dimensional neural operator. The paper gained a lot of traction and inspired many architectures which now fall under the PINN family; for a more thorough review, see [8], and for hands-on examples visit this digital book [29].

The success of this loss-based approach is apparent when considering the rapid growth of papers which extend the original iteration of the PINN. It is conceptually interpretable (though its performance pales in comparison to later methods) and can be simple to implement.

However, Krishnapriyan et al. [14] has shown that even though standard fully-connected neural networks are theoretically capable of representing any function given enough neurons and layers, a PINN may still fail to approximate a solution due to the complex loss landscapes arising from soft PDE constraints.

## 2.2 DeepONets

The DeepONet architecture is a seminal example of an infinite dimensional neural operator in contrast to the finite dimensional PINN [20]. It consists of one or multiple branch nets which encode discrete inputs to an input function space, and a single trunk net which receives the query location to evaluate the output function. The model maps from a fixed, finite dimensional grid to an infinite dimensional output space.

Since the development of the DeepONet, many novel neural operators have emerged which generalize this finite-infinite dimensional mapping to an infinite-infinite dimensional mapping [10, 12, 17, 19, 24, 26, 30], including the FNO [18]. This network operates within Fourier space and takes advantage of the convolution theorem to place the integral kernel in Fourier space as a convolutional operator.

## 2.3 Fourier Neural Operators

Concretely, the mapping between two infinite-dimensional spaces is learned from the discrete number of observed pairs. These global integral operators (implemented as Fourier space convolutional operators) are combined with local nonlinear activation functions, resulting in an architecture which is highly expressive yet computationally efficient, as well as being resolution-invariant. The FNO acts as a DeepONet with the branch net approximating the input functions and the trunk net using Fourier basis functions.

While the vanilla FNO required the input function to be defined on a grid due to its reliance on the fast Fourier transform (FFT), further work developed mesh-independent variations as well [19]. In brief, the dFNO+ and gFNO+ presented in [21] which adapt the FNO to nonlinear mappings and complex geometries respectively. Tripura and Chakraborty [30] present the wavelet neural operator (WNO) which learns the network parameters in wavelet space rather than Fourier space to allow for both frequency and spatial resolution (the latter of which Fourier basis functions are not explicitly able to capture). The WNO was shown to handle highly nonlinear PDEs with sharp changes and discontinuities on both smooth and complex geometries. The physics-informed neural operator (PINO) [19], combines the PINN methodology with the FNO resulting in an architecture which ameliorates the optimization challenges that PINNs face, as well as improving the accuracy of both models. In short, the PINO first learns a solution operator using some combination of the data loss and/or PDE loss, and then uses the learned operator as an ansatz to approximate the ground truth operator with the PDE loss (and, optionally, the operator loss from the previous step).

Neural methods are often able to operate on multiple domains, computationally efficient at test time, and can be completely data-driven. However, none of the above popular examples have a clear interpretation.

## 3 Symbolic Programming

Symbolic regression was first proposed by Koz [1] and is synonymous to regular linear or nonlinear regression except it fits over the parameters and structure of an equation rather than numerical data points. An equation can be represented as an acyclic graph, with its leaves being the operations and its terminal nodes being variables and constants. The method generates random operations as the base structure of the graph, and uses genetic programming techniques such as point mutation and crossover to introduce variation (exploration of the search space) [3]. Each equation is assessed against how many data points it can handle, often with an error metric defined using mean squared error or other standard forms. As promising equations are iterated and evaluated, the algorithm converges toward the resulting symbolic equation that best describes the data. For a more detailed overview of symbolic regression, Angelis et al. [3] presents a recent overview including the applications in machine learning to date.

Numerical models have traditionally been developed systematically by hand and with a deep under-standing of the system and its setting. Accordingly, fewer models exist for larger systems or those that are less explored. With the still-growing amount of compute available now, there is new potential for data-driven methods to discover relationships in data. Neural networks have been combined with symbolic regression in a variety of ways.

Cranmer et al. [7] capitalize on the development of graph neural networks (GNNs) [4, 28] such that the GNN has some inductive biases imposed to mimic some true latent space. This model is then trained to predict the dynamics of common physical systems, and symbolic regression is finally used to extract algebraic equations from the GNN messages and inputs. As an example, the update to some physical body's velocity may be calculated using Euler integration

$$\mathbf{v}_i^{'} = \mathbf{v}_i + \phi^v(\mathbf{v}_i, \bar{\mathbf{e}}_{\mathbf{i}}^{'}) \tag{2}$$

where $\phi^v$ is the node update function which takes the node updates $\mathbf{v}_i$ and the pooled messages $\bar{\mathbf{e}}_{\mathbf{i}}^{'}$ at the $i$-th receiver node. For the ideal case that the GNN predicts $\mathbf{v}_i^{'}$, the node update over all received and pooled messages must then be equal to the net force acting on the body. The message vectors therefore would be linear transformations of the forces, which can be extracted as symbolic expressions using symbolic regression.

## 4   The Potential for Symbolic Frameworks in Neural PDE Solvers

Some work in this direction has begun to emerge. Lee et al. [16] interprets the neural ODE by using its resulting time derivative predictions as inputs to SINDy (a framework based on sparse regression to discover parsimonious governing equations including, but not limited to PDEs) which interprets the results as a symbolic equation. In this case, two outputs are expected: the solutions across however many time steps the network predicts, and also a symbolic equation describing the former. Another similar approach by Fronk and Petzold [11] adapts the Deep Polynomial Neural Networks from [6], which constrains the output vector such that each element is represented as a polynomial based on every input element, directly for symbolic regression. Then, this network is embedded into the neural ODE to design a network capable of learning polynomial differential equations, such as those found in certain dynamical systems.

While the former two approaches add or otherwise modify an existing architecture, [2] presents a "model-of-a-model" approach called the symbolic metamodel, which takes a learned model as input and outputs a symbolic equation as a post-hoc interpretation. The symbolic metamodel is based on the Meijer G-function, $G_{m,n}^{p,q}(a_p, b_q|x)$, where $\mathbf{a}$ and $\mathbf{b}$ are two sets of real-values which specify a particular instance of $G$. One limitation is that this method cannot represent the outputs by means of any differential equations due to the limitations of the Meijer G-function. Since the target symbolic equation can be expressed (within this limited scope) as a Meijer G-function, it can be learned using standard gradient descent methods instead of symbolic regression.

However, further integrations would greatly benefit popular models like the neural operator methods described above, among others. In fact, closer interdisciplinary work involving "XAI" (explainable AI) methodology will also have bidirectional benefit. Neural solvers such as these which tackle rigorous problems such as complex PDEs pose a great stepping stone problem since it uniquely presents many ways to evaluate the correctness of an interpretation and its fitness to the network itself. Being able to verify interpretation techniques in this type of problem also makes the techniques more trustworthy to be used in less straightforward problem settings more generally.

### 4.1   Considerations for further study

Key performance indicators, such as predictive accuracy, computational efficiency, and convergence speed, need to be thoroughly assessed to determine the effectiveness and reliability of the integrated model. As in the standard neural PDE solvers, striking a balance between high-accuracy predictions and computational efficiency. Furthermore, exploring how well these models generalize across varied problem domains, scales, and complexities would offer insights into their robustness and versatility, thus making them suitable for a broader spectrum of practical applications.

Addressing the interpretability and transparency of the model's predictions stands as another cornerstone in this integration. Embedding symbolic frameworks should not only facilitate accurate and computationally efficient PDE solutions but more importantly provide clear, understandable mathematical formulations that highlight the underlying relationships and dynamics captured by the neural network. As such, the symbolic expressions derived from the model should be mathematically coherent, ideally providing a lens through which the model's predictions and decision-making processes can be analyzed and validated against known physical laws and empirical observations.

## 5    Summary

Enhanced interpretability is crucial, especially in science and engineering, where understanding the underlying processes and making sure our predictions are reliable is vital. By integrating symbolic programming with neural PDE solvers, we ensure that the models we create are not just data-driven black boxes. Instead, they have enhanced visibility thanks to their ability to respect and show physical laws and principles, filling the gap between data-driven machine learning and scientific modelling guided by theories. So, combining symbolic frameworks and neural PDE solvers opens up a promising path. It brings together the strength of machine learning and the clarity and accuracy of symbolic math expressions, pushing forward our abilities in scientific computing and simulation.

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
