# OpenReview forum: "Interpretable Neural PDE Solvers using Symbolic Frameworks"
_NeurIPS.cc/2023/Workshop/AI4Science — NeurIPS2023-AI4Science Poster_

### Official Review · Reviewer_tUDX · 2023-10-24
**A position paper discussing an important direction of research**

**Rating:** 6
**Confidence:** 4

**Review:**

The paper might be a nice contribution to the workshop w.r.t. to bringing attention to adding symbolic frameworks in PDE solvers.

---

### Official Review · Reviewer_V8p5 · 2023-10-25
**confused about the contribution**

**Rating:** 5
**Confidence:** 2

**Review:**

This paper focuses on the interpretability of neural PDE solvers using symbolic frameworks. Here are my questions.

1. It is unclear to me what is the contribution of this work. The authors propose to integrate symbolic frameworks (such as symbolic regression) into neural PDE solvers.
But they don’t clearly state how to integrate it, what are the results, and whether anyone has already tried this.
This paper has about 4.3 pages, but the authors spend only 1.5 pages discussing things beyond the introduction and existing methods.
2. I noticed a recent survey paper (Artificial intelligence for science in quantum, atomistic, and continuum systems) that also summarizes AI for PDE methods. Is there any related thing in that survey paper?
3. Typos in the fourth line of section 2 and reference [1].